# Lung Organoids for Hazard Assessment of Nanomaterials

**DOI:** 10.3390/ijms232415666

**Published:** 2022-12-10

**Authors:** Miriam T. Kastlmeier, Eva M. Guenther, Tobias Stoeger, Carola Voss

**Affiliations:** 1Institute of Lung Health and Immunity (LHI), Comprehensive Pneumology Center Munich (CPC-M), Helmholtz Center Munich, Research Center for Environmental Health, 85764 München, Germany; 2German Center of Lung Research (DZL), Comprehensive Pneumology Center Munich (CPC-M), 81377 München, Germany

**Keywords:** nanomaterial, pulmonary particle exposure, organoids, 3D in vitro models, pluripotent stem cells, respiratory toxicity, hazard assessment

## Abstract

Lung epithelial organoids for the hazard assessment of inhaled nanomaterials offer a promising improvement to in vitro culture systems used so far. Organoids grow in three-dimensional (3D) spheres and can be derived from either induced pluripotent stem cells (iPSC) or primary lung tissue stem cells from either human or mouse. In this perspective we will highlight advantages and disadvantages of traditional culture systems frequently used for testing nanomaterials and compare them to lung epithelial organoids. We also discuss the differences between tissue and iPSC-derived organoids and give an outlook in which direction the whole field could possibly go with these versatile tools.

## 1. Background

Inhalation is by far the most important route of exposure for airborne pollutants and particles. Pulmonary particle exposure comprises airborne pathogens, including viruses and bacteria, but also ambient particulate matter, such as combustion-derived particles and even engineered nanomaterials (NM); the latter mainly at occupational settings during production, processing or decomposition. Depending on their aerodynamic diameter, airborne particles bigger than a few micrometers are deposited along the surface covered with mucus of the conducting airways by impaction, where they are rapidly removed via mucociliary clearance. Inhaled nanoparticles (NP) smaller than 100 nm in diameter deposit mainly by diffusion in the whole lung, but are especially efficient in the most distal and fragile parts of the lung, the alveoli [1]. While the alveolar region possesses over 90% of the lung’s surface area, it also represents the most susceptible tissue interface to the environment with only a few 100 nm thickness of the alveolar walls, protected only by a thin liquid layer [2]. The primary interaction during inhalation of particles occurs, therefore, with either mucus covering the conducting airways or alveolar lining fluid of the respiratory tract. Pulmonary surfactant as the major component of the lining fluid, consists of a unique composition of 80–90% phospholipids, 5–10% neutral lipids and 10% surfactant-associated proteins (SP-A, B, C and D) [3]. The surfactant acts as a surface tension lowering film covering the alveolar surface, thereby protecting the alveoli from collapse during exhalation and reduces the effort of breathing [2]. In addition, any deposited material or particle is immersed into the lining fluid. The interaction between lining fluid and particles may also dramatically change the physical–chemical properties of alveolar deposited inhaled particles, causing immobilization or aggregation, and modifies their surface chemistry. Particle clearance is facilitated by either removal via the mucociliary escalator in conducting upper airways or phagocytosis by alveolar macrophages (AMs) roaming the alveolar surface. Ineffective clearance, repetitive inhalation as well as hotspots of deposition formed at the bifurcations of terminal bronchioles and alveolar ducts, can lead to accumulation and high particle burden at specific areas of the respiratory tissue, and may thus increase the per cell delivered dose dramatically [4]. Furthermore, and in dependence of particle chemistry, its deposition may damage the surfactant function of the layer itself [5] and lead to a local inflammation [6].

Once reaching the alveolar surface, particles can lead to serious health consequences such as attenuated lung development for children exposed to combustion-derived traffic emissions [7,8], cardiovascular effects in susceptible adults as for diesel exhaust particles [9] and metal fume and polymer fume fever as for specific metal oxides and fluorinated polymers [10]. Depending on the pulmonary delivered dose, basically all materials can cause local inflammatory responses, in this context a variety of toxicological rodent studies support the respiratory toxicity of particles with particle surface area as the most valuable predictor for acute lung inflammation [11]. Detrimental long-term consequences including chronic inflammation, fibrosis and even tumor formation in lung tissue have been associated with inhalation of certain types of fiber shaped, high aspect ratio NPs [12]. Despite this knowledge, the ever-growing field of nanotechnology-associated nanomaterial toxicology requires smarter approaches for NM fabrication, grouping and testing, especially considering high throughput approaches, ethical commitment and at the same time replacing and reducing animal testing [13].

To achieve a smarter and more ethical approach to NM testing, the Adverse Outcome Pathway (AOP) framework has been established, which incorporates mechanistic knowledge generated from in vivo experiments to connect measured toxicological endpoints with a pathological consequence by a sequence of molecular initiating events (MIEs), consecutive key events (KEs) and the final adverse outcome (“disease”; AO). Several AOPs have been identified and shown to have strong correlation across published in vivo datasets [14]. To get robust information about the connection of KEs, AOPs need to particularly assess quantitative relationships, e.g., relevant NM doses. Furthermore, this AOP approach facilitates the design of superior in vitro testing strategies with the ultimate goal to reflect MIEs or KEs robustly in vitro, which would ultimately unburden safe-by-design strategies and reduce animal testing in the future. Recently, for the AOP ‘chronic inflammation’, an in vitro based test system has been demonstrated with highly specialized methods to reach superior predictive power for an ample set of NMs (metal oxide-based materials) [14]. AOPs are especially helpful for deciding which New Approach Methodology (NAM) could be used regarding NM toxicology studies [15]. With appropriate NAMs, toxicity testing is evidence-based, more predictive and reproducible. Hence, more and more predictive alternative and tissue specific in vitro models have to emerge based on AOPs. These will enable reliable and high throughput applicable cell-based studies, covering information from the molecular onset to the development of pathology, namely the identification of MIEs and KEs leading to AOPs in vivo.

In the following section, we will (1) portrait the difficulties of current in vitro models especially for specified AOP based testing, (2) introduce different lung organoid cultures as an alternative method and (3) give an outlook on these NAMs in the field of research.

## 2. Culture Methods for NM Hazard Assessment

Numerous studies display adverse effects of NM on the lung or lung cells, including cell proliferation, oxidative stress [16,17], DNA damage [18], pro-inflammatory [19,20] and pro-fibrotic response [21,22] using in vitro or in vivo systems to detect and compare molecular effects of different NMs, and to identify potential detrimental responses through nanoparticle-specific actions. For an in vitro set up, the standard and most simple technique in toxicological research is achieved by adding substances directly to the media of submerged cultures. However, for inhalation and particle toxicological studies this method is not decisive, since the process of particle–cell interaction as observed at the epithelial surface of the lung, is different to in medium submerged conditions [23,24]. The distribution pattern of NPs by inhalation is more critical than the stimulation itself [25]. Apart from inappropriate biological conditions, obscure dosimetry, especially the dose interacting with and thus delivered to the cell at submerged conditions, is a major concern for poorly soluble particles that is challenging to determine and, moreover, is still rarely considered [24,26]. The unrealistic dose delivery for the lung surface is mainly due to factors driving the sedimentation route in submerged cultures. For example, the aggregation of NMs in serum protein containing media, or the possibility of dissolution of certain NMs in high volumes of media can result in an unrealistic distribution of particles across the exposed cells [27,28].

To overcome these disadvantages for inhaled particles, cells can be cultured on an Air–Liquid Interface (ALI). By placing the cells or tissue on a porous membrane and feeding them just from the basal side, the apical side is open for an inhalation like airborne exposure, thereby a comparable experimental set up to in vivo conditions arises. Hence, in vitro exposure at the ALI with airborne NMs is not only the more realistic approach, but also the one allowing defined cell delivered dose estimations compared to exposure under submerged conditions. ALI inhalation models have the potential for a more precise reproduction of the processes during exposure, as they can mimic the fragile respiratory epithelial region comparable to structural in vivo terms [29]. Especially for studying the effects of exposure to low solubility materials, a special Air–Liquid Interface cell exposure (ALICE) system was developed, which uses a nebulizer to generate a droplet cloud of dispersed particles. Then, in the exposure chamber, the created moisture cloud finally drives the applied NMs to gravimetrically deposit onto the culture [28]. Instead of the use of gravimetric force, which requires aqueous dispersion for nebulization, other methods use electrostatic force to improve the deposition efficiency on the ALI surface [30]. Alternatively, ALI cultured cells can be exposed by using continuous flow systems (CFSs), which offer more realistic dose rates. CFSs may be especially advantageous where the cell exposure shall get directly linked downstream of particle emission or production [31]. In this context it must be mentioned that the exposed cells are often immortalized cell lines, which may resemble the natural cell characteristics only partly. In recent years, primary cells have been increasingly used to recapitulate physiological features in a feasible manner. In addition, the porous membranes used as substrate for the cell medium interface usually exceed realistic dimensions. Notably, well-working approaches to overcome this problem with advanced biomimetic membranes already exist [32].

Even with the most desired advanced models, it is noted that the results generated by inhalation of nanoparticles in vivo cannot be fully and properly represented in vitro. Previous studies have shown that the use of immortalized cell lines does not represent the in vivo situation completely, so does not provide fully comparable results to those obtained in vivo. This relates to the fact that immortalized cell lines often lose polarity and lack key morphology features, which may biologically distinguish respective cells in the context of tissue. Furthermore, as the immortalized cells do not have a natural proliferation cycle due to mutation or manipulation, they have evaded normal cellular senescence and instead can keep undergoing division, which could lead to functional alterations and genetic drifts [33,34]. In general, any cell model will only model a certain biological aspect of the in vivo situation and this aspect and its limitations have to be well-known to the researcher to use the model appropriately. Several human alveolar epithelial cell lines, for example A549, NCI-H441, TT1 and hAELVi, are commercially available. The ones originating from alveolar type 2 cells (AT2s) mostly lost their stem cell character, referring to the possibility to differentiate into alveolar type 1 like-cells (AT1s), with protein expression of Aquaporin-5 (AQP5) or Podoplanin (PDPN) [35,36,37] as it occurs in the lung. TT1 and hAELVi represent cells with an AT1-like phenotype regarding morphology and caveolae presence, although they do not display other common AT1 markers like AQP5 or, in the case of TT1, only show discontinuous tight junctions [38,39,40,41]. To get a human epithelial cell line representing the bronchial epithelium, for example BEAS-2B, 16HBE14o or Calu-3 are well established [42]. Indeed, there are also murine lung epithelial cell lines, namely MLE-12 or LA-4, representing the alveolar compartment. Therefore, the lack of reproducibility between in vivo and in vitro data is not due to the applied in vitro model, but rather to the cells chosen for the particular research aim.

A promising approach to overcome disadvantages of currently widely used immortalized cell lines and to compare results created in vivo with in vitro data is the use of three-dimensional (3D) cell cultures, the so-called lung organoids. Organoids are defined as three-dimensional, mostly spherical shaped constructs cultured in vitro in an extracellular matrix. They self-organize from single stem cells into multicellular structures and mimic the in vivo organ, in this case the bronchiolar or alveolar region of the lung [43]. An overview of different ways to generate lung organoids and their cells of origin is shown in Figure 1.

One method to grow lung organoids is to isolate primary epithelial cells out of lung tissue. This is possible with murine lungs as well as human tissue, although the availability of human lung tissue is limited. Basal cells act as progenitor cells in the tracheal and bronchial region of the lungs [54]. When isolated and cultured in a complex matrix, airway basal cells can form bronchospheres and contain multiple airway cell types, including ciliated, goblet and secretory cells, with expression of markers as Forkhead Box J1 (FoxJ1), acetylated a-tubulin, Mucin 5AC (MUC5AC), Cystic Fibrosis Transmembrane Conductance Regulator (CFTR) or secretoglobin family 1A member 1 (SCGB1A1). The human and murine bronchospheres still contain basal cells expressing for example p63, enabling them to self-renew [44,45]. In the alveolar region, AT2s have stem cell character and can proliferate and differentiate into AT1s [48]. To obtain organoids, mesenchymal support cells are often needed to help the organoids grow. Human mature alveolar organoids show AT2 markers such as surfactant protein-C (SFTPC) and HTII-280. Murine alveolar organoids also contain SFTPC expressing cells and, in addition, cells showing AT1 characteristics [48,49] (Figure 2a,b). Thus, stem cell properties are retained within a 3D culture, in contrast to traditional culture methods with cell lines. In addition to these two organoid types, the bronchospheres and the alveolar organoids, it is also possible to obtain bronchioalveolar organoids from distinct cell populations in mouse lungs. The so-called bronchioalveolar stem cells (BASCs) and Scgb1a1 positive club cells are able to give rise to organoids containing cells with an airway phenotype as well as alveolar characteristics. They combine both lung compartments in vitro, with bronchiolar cells in the center followed by an outer part of branching alveolar structures [46,47].

An alternative to primary lung epithelial cells for generating lung organoids is the use of directed differentiation of induced pluripotent stem cells (iPSCs). Since the discovery of human iPSCs [55], they are considered a valuable alternative to the problematic use of embryonic stem cells (ESCs) and to provide comparable in vitro models in relation to the actual disease pattern in humans with the potential of long term and repetitive experiments.

The experimental set-ups of in vitro lung models are based on biochemical differentiation of hiPSCs into lung lineages. Organoids derived from stem cells (ESC or hiPSC) are able to differentiate and self-organize through lineage bonding comparable to processes taking place during development in vivo [56].

hiPSCs differentiated into lung progenitors can be used for deriving airway organoids. They contain SCGB1A1+ secretory cells, multiciliated cells expressing FOXJ1 and basal cells, amongst others [50]. In modified conditions, lung progenitors can grow into mature alveolar epithelium with specific cell expression markers of AT2s (and AT1s), e.g., SP-C [51,52]. As shown in Jacob et al. 2017 [51], NKX2.1 is highly expressed in tightly packed lung progenitor colonies. At a later stage of differentiation, lung progenitors resulted in self-renewal and high yield of SP-C expressing iAT2s (Figure 2c,d). An interesting approach to obtain lung organoids that contain AT2s, AT1s as well as airway goblet cells, is to generate lung bud organoids by prolonged differentiation in a 3D matrix. With this method, mesenchymal cells expressing Vimentin (VIM) arise, surrounding the organoids [53].

Great advantages of 3D lung organoid cultures compared to conventional cell lines are the comparable cellular identity and functionality to the in vivo situation, and the potential to differentiate into several epithelial cell types. This enables us to perform disease modeling, developmental and regeneration studies, identify roles of the distinct cell types regarding cellular communication in defined settings and create a representative model of airway and/or alveolar lung compartments. When comparing architecture and functional readouts of lung tissue, a 3D cell culture system creates much better and even more realistic conditions than a cell monolayer culture system [57,58]. A feature of mature AT2s in a 3D cell culture system, is the ability to produce lamellar body-like inclusions, including mature SP-B and SP-C protein forms, and thus further supporting their self-renewing capacity, which is desperately needed for a constant repetition of experimental set-ups. Lipidomic analysis of the intracellular and extracellular material from alveolar organoids show amounts of dipalmitoylphosphatidylcholine (DPPC), the main phospholipid in surfactant, and thus the presence of functional lamellar bodies that synthesize and secrete surfactant from phenotypically mature AT2s [51]. At the moment, this prominent feature of AT2s is only found in stem cell derived 3D cultures. Another advantage of organoid cultures, either originating from primary lung cells or iPSCs, is the possibility to include multiple defined cell types into a co-culture system. The defined, but superior model can incorporate different cells representing lung epithelial cells interacting with fibroblasts, endothelial cells or immune cells [46,59,60,61,62], thus promoting interactions and displaying inflammation and cell-matrix alterations, for example. Especially, studying cell–cell interactions with regard to therapeutic efficacy and toxicity of delivered drugs is possible in 3D microtissue models. One thing to highlight as an advantage of human organoid cultures is they provide faster and more robust outcomes, as well as a more accurate representation of human tissue than animal models [63]. Notably, from hiPSCs, generated lung organoids can be passaged for up to 300 days and retain their typical alveolar characteristics [52,64]. In contrast to all of these advantages of using iPSC derived organoids, their generation is quite laborious. For human tissue derived lung organoids, the availability of lung samples to perform epithelial cell isolation is restricted, and obvious ethical issues arise in this context. An additional dilemma regarding human lung tissue samples is that it is not feasible to get completely healthy tissue, only, for example, peritumoral samples. Although murine lung organoids can be derived from various genetic backgrounds, this method is still dependent on animal experiments and is not the replacement that traditional culture models are. Nevertheless, murine as well as human organoid experiments could help to reduce the number of research animals used in accordance to the 3R principles [65] and additionally, using human cells would increase the translational aspect and allow patient-associated studies. Especially in the context of NM toxicity assessments, it is advisable to take advantage of the benefits 3D organoids offer. Lung organoids are already used for different research questions regarding NM toxicity. Readouts including reactive oxygen species (ROS) production, epithelial cell differentiation and regeneration, NP internalization or surfactant production can be assessed easily, and help to elucidate the mechanisms underlying disease progression in the lung after NP exposure [66,67,68]. Toxicity testing in organoids is not yet used often, but these examples already show the numerous opportunities with 3D lung cultures. However, one difficulty still is to imitate the inhalation of NMs. For example, in Yu et al. 2022 [68], the particles to be tested are mixed into the culture medium, which is without a doubt a convenient and high throughput suitable approach for NM exposure, but leads to similar problems regarding the cell delivered dose and the particle–cell interaction as a conventional 2D submerged cell culture does. Nevertheless, 3D organoids are able to respond to stimuli and can recapitulate epithelial cell responses more accurately than 2D culture [69]. In addition, usually grown alveolar organoids are polarized in such a way that the surfactant producing apical side is faced towards the lumen of the sphere. Thus, exposure to NMs through the media or matrix does not reach the epithelial cells as it would in vivo, which are exposed from the basal side. One possible idea to overcome this issue is to microinject the desired harmful substance directly into the lumen of the organoid, which is not yet performed with NMs, but within several other contexts [59,70]. This brings the NMs or pathogens directly to the site of action and the exact dose delivered to the cells is known. Nevertheless, microinjection of NM into lung organoids is not done so far as it is challenging to generate high throughput. In order to get a relevant outcome, this method requires experience regarding the microinjector. On the other hand, there are already approaches to change the polarity of distal lung organoids towards an apical-out polarization [71] This method could be used to expose lung organoids to NMs easier from the apical cell side. Still, organoids are grown in matrix with feeding medium, this means a direct contact or defined cell delivered dose is difficult to achieve under these conditions.

Another approach to take advantage of the stem cell character that cells keep in organoid culture is to dissociate the cultured 3D organoids into single cells again. When cultured in Transwell inserts, organoid derived epithelial cells can form an intact epithelial barrier [72].

In this setting, exposure to NMs using the ALICE system, where particles are nebulized and a defined dose is distributed equally upon the cells, is feasible. The combination of using cells with functions and properties as in vivo, and the inhalation-like exposure to particles with the ALICE system makes this culture method interesting. In summary, the combination of organoid culture and subsequent ALI exposure to balance the limitations of each individual model will be a useful approach to assess NM hazards.

## 3. Future Direction

Lung organoid technology has developed quickly in the last years and became a useful tool for modeling perpetuating lung diseases and hazards affecting the lung [72]. With reference to previous research, it is evident that a holistic in vitro model of the lung cannot be generated. Therefore, it is absolutely essential for a comprehensive, accurate and above all realistic test result to relate the model to the specific research question. It must be clarified from the beginning whether a 2D submerged, ALI model or a 3D cell culture model would be the right choice for the problem posed. For investigations, particularly with regard to epithelial responses, epithelial cell differentiation and epithelial recovery, organoids are a suitable instrument [72].

Thus, the choice of cells used should be thoroughly considered, especially concerning their respective properties, such as forming lamellar bodies, producing surfactant or retaining stem cell character. It becomes clear that there is not one overall cell line for a general experimental setup, for instance with regard to NM inhalation, where particle–cell interactions in a realistic environment are of particular importance. Therefore, using advanced target cells that are able to create a liquid lining layer would improve the comparability of in vitro studies to in vivo findings and lead to extended outcomes (Table 1).

One important step in the future is to increase the use of stem cell derived murine or, even better, human cells that adequately reflect the disease pattern for monitoring and understanding the underlying cell–cell interactions after NM exposure. For instance, the use of immortalized cell lines within an in vitro experiment has shown to be not comparable to a clinical picture. Isolated human primary cells can only be passaged for a short period of time and are therefore also not sufficiently suitable for a complex experimental set-up with necessary replicates. At this point, an adapted experimental setup with hiPSCs would be a desirable and new promising approach. Due to their close resemblance to the primary cells, despite their durability and the possibility to be passaged over a long period, hiPSC derived organoids should be the prospective choice for human in vitro experiments.

In addition, an adequate murine in vitro 3D culture system has several advantages. It is important to create setups reflecting and confirming the findings observed in previous in vivo studies. This enables us to elucidate cell–cell interactions and events happening on cellular, protein and gene levels, while reducing the number of animals used in similar in vivo testing according to the 3R principles. Based on the AOP framework regarding NM toxicity, lung organoid culture could be a helpful NAM to obtain results representing in vivo conditions more accurately. With the emergence of new analytical techniques, profiling cellular responses at the single cell level, we realize that a tissue such as the lung consists of over 50 different cell types [73]. Yet these new approaches, such as single cell transcriptomics, raise the awareness that very specific cellular niches might be required to sense injury. For example, an AOP initiating event caused by inhaled particles, and a distinct cell–cell communication network are then required to develop the pathological outcome. For the lungs, these cellular networks and outcomes are now increasingly described for SARS-CoV-2 infection and pulmonary fibrosis [74], but similar communications are likely required for nanoparticle triggered AOPs. Reproducing the underlying key events and cell interactions at the in vitro level will be of great impact for future safety testing and organoids because of the maintained cellular plasticity and more natural cellular communications hold great promise.

In summary, we illustrate that already established experimental setups with new and adapted cells will lead to potentially improved or even new results and findings. Lung organoids include these particular cells, enabling us to perform hazard assessments for NMs within suitable models.

## Figures and Tables

**Figure 1 ijms-23-15666-f001:**
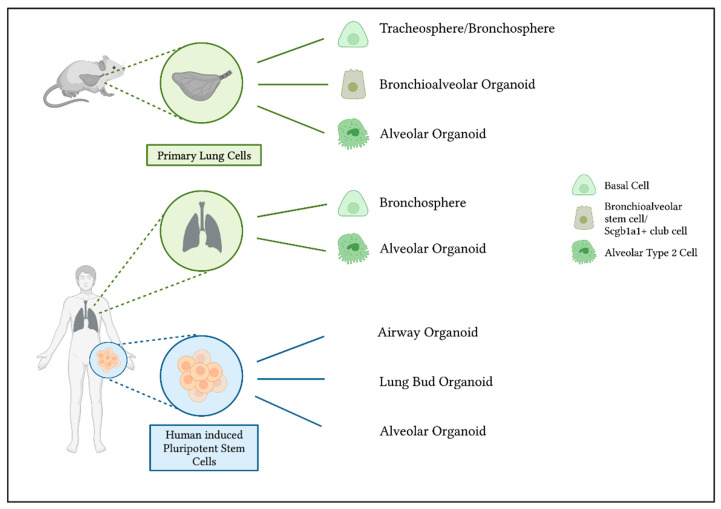
Generation of murine and human lung organoids and their cells of origin. Organoids can be derived from primary murine and human lung cells. Tracheospheres and bronchospheres originate from airway basal cells [44,45]. To generate bronchioalveolar organoids from murine lungs, bronchioalveolar stem cells or Scgb1a1+ club cells can act as progenitors to bronchiolar as well as alveolar cells [46,47]. Primary isolated alveolar type 2 cells are able to differentiate into alveolar organoids [48,49]. Another possibility to generate lung organoids is the use of human induced pluripotent stem cells. The use of different growth factors and conditions results in either airway [50] or alveolar organoids [51,52]. Organoids that include bronchial as well as alveolar cells can be derived as so-called lung bud organoids [53].

**Figure 2 ijms-23-15666-f002:**
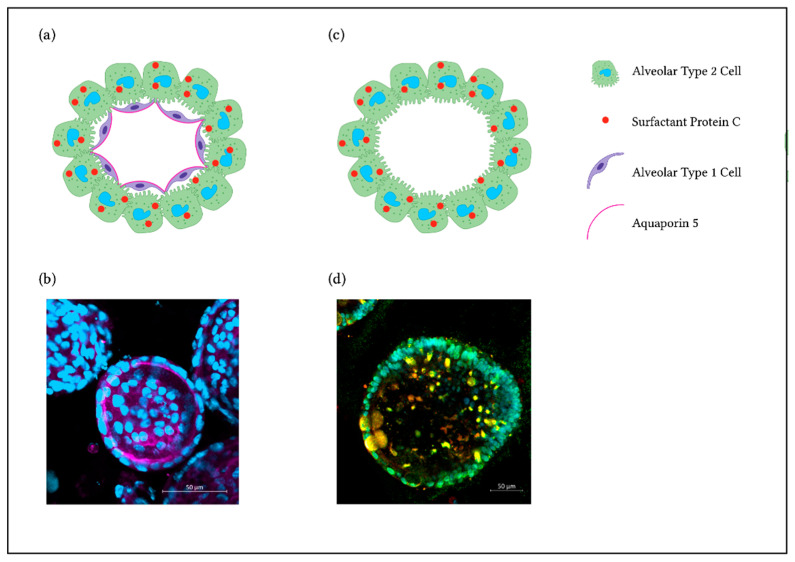
3D Alveolar Organoids. (**a**) Illustration showing a primary murine alveolar organoid. (**b**) Representative immunofluorescence staining of a murine alveolar organoid showing AQP5 staining as a marker for AT1s (pink) and nuclei (DAPI, blue). (**c**) Illustration showing a hiPSC-derived alveolar organoid. (**d**) Representative immunofluorescence staining with AT2s expressing SP-C (red), NKX2.1 (green) and nuclei (DAPI, cyan).

**Table 1 ijms-23-15666-t001:** Comparison of cell lines and organoids and exposure recommendations.

		Accessibility	Feasibility	Physiological Characteristics	Represented Cell Types	Co-Culture	Exposure Methods for Hazard Assessment			
							Submerged	ALI	CFS	Microinjection
**Cell Lines**		Commercially available, many passages	Easy to maintain	Partially preserved	Single	2D layered structure, often with use of membranes possible	Easy to apply, cell-delivered dose challenging to determine, HTS	Mimics deposition of inhaled particles, defined cell-delivered dose, realistic nano-bio interphase (surfactant etc.)	Mimics deposition of inhaled particles with realistic dose rate, defined cell-delivered dose, realistic nano-bio interphase (surfactant etc)	3D structure required
**Primary Cells**		Animal or human tissue required, limited passaging	Isolation expertise required	Partially preserved	Single	2D layered structure, often with use of membranes possible	Easy to apply, cell-delivered dose challenging to determine, improved IVIVC			
**Organoids**	**Primary Cell-Derived**	Animal or human tissue required, limited passaging	Isolation expertise required	Mostly preserved	Formation into organoids containing AT2s, AT1s and airway epithelial organoids in the same culture	Organotypic, 3D self-assembly, possible	Easy to apply, cell-delivered dose challenging to determine, improved IVIVC, HTS exposure from basal instead of apical side	2D structure and ALI culture required		Delivers NM directly to apical side within the organoid lumen, high IVIVC, technologically challenging
	**hiPSC-derived**	Long-time passaging of organoids	Complex differentiation procedure, high level of organoid maintenance	Comparable to in vivo	Differentiation into organoids containing AT2s, (AT1s) or airway organoids	Organotypic, 3D self-assembly, possible				

## Data Availability

Data sharing not applicable.

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
