# Peer review of "Lung Organoids for Hazard Assessment of Nanomaterials"

_ijms, 2022, doi:10.3390/ijms232415666_

Round 1

Reviewer 1 Report

- The authors can also mention that particles <100 nm deposit by diffusion and hence also in the upper respiratory tract, specifically in the nose

- AOPs are not the holy grail, they need to be made quantitative to understand how much of a substance in needed to go from one key event to another, this should be mentioned.

- cell proliferation is a very important response that can potentially lead to cancer or tissue remodeling. This does not necessarily require oxidative stress, and can be inserted in ln 86

- ALI (ALICE) systems may result in a more relevant exposure and better understanding of the dose (yet still very high dose rate), but continuous flow systems are not mentioned

- I think that many research groups are now also using primary cell cultures so the statement "exposed cells are often immortalized cell lines" ln 119 may not be up to date.

Table 1 does not fit on the page so cannot be reviewed

It would be given the paper some added value if the reader can also read the recommendations in a table: when to use which model and exposure set up

Author Response

Reviewer #1

Comments and Suggestions for Authors

- The authors can also mention that particles <100 nm deposit by diffusion and hence also in the upper respiratory tract, specifically in the nose

We thank the reviewer for highlighting this issue. In line 29, we included, that particles <100 nm deposit by diffusion therefore affecting the whole lung.

“Inhaled nanoparticles (NP) smaller than 100 nm in diameter deposit mainly by diffusion in the whole lung but especially efficient in the most distal and fragile parts of the lung, the alveoli.[1]”

- AOPs are not the holy grail, they need to be made quantitative to understand how much of a substance is needed to go from one key event to another, this should be mentioned.

Mentioning quantitative AOPs is really important in the field of nanomaterial toxicity, we included this viewpoint now in our Perspective (l. 72)

“To get robust information about the connection of KEs, AOPs need to assess particularly quantitative relationships, e.g. relevant NM doses.”

- Cell proliferation is a very important response that can potentially lead to cancer or tissue remodeling. This does not necessarily require oxidative stress, and can be inserted in ln 86

We thank the reviewer for this comment. We have included cell proliferation as important readout in line 91.

“Numerous studies display adverse effects of NM on the lung or lung cells, including cell proliferation, oxidative stress [16,17], DNA damage [18], pro-inflammatory [19,20] and pro-fibrotic response [21,22] using in vitro or in vivo systems to detect and compare molecular effects of different NMs and to identify potential detrimental responses through nanoparticle-specific actions.”

- ALI (ALICE) systems may result in a more relevant exposure and better understanding of the dose (yet still very high dose rate), but continuous flow systems are not mentioned

We thank the reviewer for highlighting the missing technology. We added a comment in line 122.

“Alternatively, ALI cultured cells can be exposed by using continuous flow systems (CFSs), which offer more realistic dose rates. CFSs may be especially advantageous where the cell exposure shall get directly linked downstream of particle emission or production [31].”

- I think that many research groups are now also using primary cell cultures so the statement "exposed cells are often immortalized cell lines" ln 119 may not be up to date.

The reviewer is absolutely right, we added the emerging use of primary cells nowadays.

Line 128

“In recent years, primary cells have been increasingly used to recapitulate physiological features in a feasible manner.”

- Table 1 does not fit on the page so cannot be reviewed

The table is now clearly readable and some information is added targeting the next point.

- It would be given the paper some added value if the reader can also read the recommendations in a table: when to use which model and exposure set up

We agree to the reviewer`s idea and added respective information into existing Table 1 to provide a summary of the different in vitro exposure models comparing cell lines and organoids. Of course, the challenge is the particular research question that needs to find agreement with the level of complexity of the model used (e.g. cell lines 2D, submerged, ALI, primary cells, 3D, stem cell derived, human or rodent). We have added more detailed information about possible applications to the table to guide researchers but we are aware that choices will require individual consideration and trade-off for each research question.

Reviewer 2 Report

The review describes in vitro cell models that may be used for realistic risk assessment of particle i.e nanoparticles. The review lists a number of well established in vitro pulmonary cellular and exposure systemes for the purpose of their applicability in New Appeoach Methodologies. Lung organoids generated from human lung or rodent tissues or iPSCs are highlighted as promising models.

The model is appropriate however, the authors should discuss how multicellular interactions in the air-blood barrier i.e epithelial-immunecell-endothelial could be studied in the organoids or will it be a weakness of the organoids?    

Author Response

Reviewer #2

Comments and Suggestions for Authors

The review describes in vitro cell models that may be used for realistic risk assessment of particle i.e nanoparticles. The review lists a number of well established in vitro pulmonary cellular and exposure systems for the purpose of their applicability in New Approach Methodologies. Lung organoids generated from human lung or rodent tissues or iPSCs are highlighted as promising models.

The model is appropriate however, the authors should discuss how multicellular interactions in the air-blood barrier i.e epithelial-immune cell-endothelial could be studied in the organoids or will it be a weakness of the organoids?    

Thanks for this important suggestion. In line 244 we mentioned, that organoid cultures can include different cell types in order to investigate cell-cell interactions. We have added your valuable input about the mentioning of the endothelial compartment and have changed from macrophages to “immune cells” to show the open possibility to include a wide range of immune cell types.

Please refer to this section, starting in line 240:

“Another advantage of organoid cultures, either originating from primary lung cells or iPSCs, is the possibility to include multiple defined cell types into a co-culture system. The defined, but superior model can incorporate different cells representing lung epithelial cells interacting with fibroblasts, endothelial cells or immune cells [45,58-61], thus promoting interactions and display inflammation and cell-matrix alterations for example. Especially studying cell-cell interactions with regard to therapeutic efficacy and toxicity of delivered drugs is possible in 3D microtissue models.”